# Biological Approach in the Treatment of External Popliteal Sciatic Nerve (Epsn) Neurological Injury: Review

**DOI:** 10.3390/jcm11102804

**Published:** 2022-05-16

**Authors:** Alejandro León-Andrino, David C. Noriega, Juan P. Lapuente, Daniel Pérez-Valdecantos, Alberto Caballero-García, Azael J. Herrero, Alfredo Córdova

**Affiliations:** 1Department of Orthopedic Surgery, Clinic University Hospital of Valladolid, 47005 Valladolid, Spain; aaleon@saludcastillayleon.es; 2Department of Surgery, Ophthalmology, Otorhinolaryngology and Physiotherapy, Faculty of Medicine, University of Valladolid, 47005 Valladolid, Spain; 3SCO (Scientific Chief Officer) Laboratorio de Biología Molecular y Celular R4T, University Hospital of Fuenlabrada, 28942 Fuenlabrada, Spain; jplapuente@yahoo.es; 4Biochemistry, Molecular Biology and Physiology, Faculty of Health Sciences, GIR Physical Exercise and Aging, University of Valladolid, Campus Duques de Soria, 42004 Soria, Spain; danielperezvaldecantos@gmail.com (D.P.-V.); a.cordova@uva.es (A.C.); 5Department of Anatomy and Radiology, Faculty of Health Sciences, GIR Physical Exercise and Aging, University of Valladolid, Campus Duques de Soria, 42004 Soria, Spain; alberto.caballero@uva.es; 6Department of Health Sciences, Miguel de Cervantes European University, 47012 Valladolid, Spain; jaherrero@uemc.es

**Keywords:** external popliteal sciatic nerve (EPSN), common peroneal nerve, compression, neurolysis, foot drop, growth factors

## Abstract

The external popliteal sciatic nerve (EPSN) is the nerve of the lower extremity most frequently affected by compressive etiology. Its superficial and sinuous anatomical course is closely related to other rigid anatomical structures and has an important dynamic neural component. Therefore, this circumstance means that this nerve is exposed to multiple causes of compressive etiology. Despite this fact, there are few publications with extensive case studies dealing with treatment. In this review, we propose to carry out a narrative review of the neuropathy of the EPSN, including an anatomical reminder, its clinical presentation and diagnosis, as well as its surgical and biological approach. The most novel aspect we propose is the review of the possible role of biological factors in the reversal of this situation.

## 1. Introduction

Traumatic peripheral nerve injury is a difficult and controversial issue for the orthopaedic surgeon and a challenge for rehabilitation. Despite the introduction of microsurgical techniques by Kurze [1], nerve repair and functional recovery is mostly incomplete and always difficult to predict. Moreover, nerve injuries remain the main causes of reduced functional capacity and generate high socio-economic costs due to the long rehabilitation times required, as well as the disability sequelae that may eventually result [2,3].

Fibular or peroneal neuropathy is the most common lower limb neuropathy and the third most common focal neuropathy found in general, after median and ulnar neuropathies [4]. Following high tibial and fibular osteotomies, an incidence of peroneal neuropathy has been observed in 2–27% of patients [4]. Following knee dislocations, common peroneal nerve injury has been observed in 16–40% of patients. In children, peroneal neuropathy of the common peroneal nerve was also observed to be affected most frequently (59%), followed by the deep (12%) and superficial (5%) peroneal nerves [5].

To the best of our current knowledge about nerve trauma and neuronal regeneration, the solution to improve the outcome of peripheral nerve repair is biological rather than surgical or rehabilitative [3,6,7]. Progress can only come from understanding and being able to modulate the different biological phases involved in the repair of peripheral nerve injuries, as there are phenomena of nerve regeneration fatigue, fascicle mismatches, and effector degeneration [8].

The mechanisms of injury are mainly contusions, compression, traction, focal ischaemia, and total or partial section. In practice, all degrees of involvement are observed, from conduction blocks to neurapraxia, axonotmesis, and neurotmesis. The most frequent causes of nerve involvement in the lower extremity are: penetrating trauma; fractures; dislocations; and iatrogenesis during injection or surgery, especially total knee arthroplasty [2,9,10,11]. The pathogenesis of these lesions progress in complexity from neurapraxia (a punctual conduction block due to myelin damage, as in compressive neuropathies) to axonotmesis (an axonal injury, due to crushing or traction, with irreversible damage associated with denervation time of the target muscle, but with a favourable prognosis of the nerve) and to neurotmesis (a complete section of the nerve with destruction of the endoneurial tubes that requires surgical treatment for resolution, and appears in penetrating wounds or ischemic processes) [11].

Microscopic techniques have shown that nerve morphology is normal and neuromuscular junctions are maintained in chronic compression lesions. However, the myelin sheath is thinner and degraded, and there is decreased internodal length (the distance between adjacent nodes of Ranvier) [12].

The main mechanical characteristic of the peripheral nerve is the tensile strength with a non-linear behaviour between weight and deformation. Under constant elongation, the nerve tension is reduced to 30% in the first 10 min and very little more in the next 20 min. This relaxation phenomenon (creep) is useful in sutures and nerve grafts, as the remaining tension will be lower after a short time [13].

Regarding the mechanisms thought to lead to compression injuries, from an anatomical point of view, the narrowing of the openings causes an increase in pressure at that site, compresses the blood vessels, and leads to nerve ischaemia. Another proposed mechanism is that, as a result of lower pressure, which decreases venous return, it can lead to venous stasis. Over time, this situation can lead to extraneural oedema, with a consequent increase in fibrous tissue around the nerve [14].

Not all cases resolve favourably because up to 33% of cases in peripheral nerve injuries have incomplete recovery with poor functional outcomes. Partial recovery has been observed, sometimes with a complete loss of motor and sensory function, and with chronic pain and muscle atrophy [1,15,16]. Compression of the EPSN is associated with peripheral neuropathy [17].

Nowadays, more and more importance is being given to the role that biological therapy can play after surgery for peripheral nerve disorders. This paper reviews the general aspects of surgery in traumatic compression of the EPSN. Special emphasis is placed on the possibilities of biological treatment for nerve regeneration after surgery.

To conduct the narrative review, a comprehensive literature search was performed using PubMed, Ovid MEDLINE, and EMBASE databases and the following search terms: (“nerve” OR “nerve trauma” OR “neurological surgery” OR “peripheral nerve injuries” OR “nerve repair” OR “nerve regeneration” OR “paralysis of the external popliteal” OR “compression and entrapment neuropathy” OR “fibular nerve compression” OR “peroneal nerve” OR “repair Schwann cell” OR “neurotrophic factors” OR “nerve regeneration” OR “mesenchymal stem cells” OR “fibroblast growth factors” OR “adipose stem cells” OR “platelet-derived growth factors” OR “myelination”. We first selected the articles based on what we found in the abstracts, which led to a more exhaustive selection by reading the selected articles. We must take into account that many of them are very similar and do not contribute more than other articles.

## 2. Anatomy of the EPSN

The EPSN is the external division branch of the sciatic nerve. As it passes through the thigh, it is responsible for the innervation of the short head of the biceps femoris. From its origin, it descends outwards, following the biceps cruris tendon, and then it branches off and becomes independent of the sciatic nerve at the level of the popliteal fossa (Figure 1). The global function of the EPSN is dorsiflexion of the ankle at the moment of foot stance, directing the toe outward [18,19].

## 3. Compression of the EPSN

There are several causes of nerve injury, the most frequent being the compressive cause; however, in this review, we do not wish to expand on this aspect. Schematically, the etiology may be due to: sustained nerve compression; trauma; peripheral neuropathies; very strenuous exercise; viral infection; and idiopathy [20,21,22,23,24]. From a pathophysiological point of view, nerve compression causes alterations in the intraneural blood microcirculation and axonal lesions, and alterations in the supporting connective tissue, among others [16,21]. These alterations, maintained over time, lead to demyelination, conduction disorders, and degeneration of nerve fibres [8,19,25,26,27].

Endoneural oedema increases hydrostatic pressure leading to endothelial hypoxia and consequent axonal damage [28,29]. Segmental axonal ischaemia is produced by a decrease in blood flow involving a loss of energy for transport and dysfunction of the sodium pump system. The cell membrane is also affected by the energy failure and has a consequent loss of conduction and transmission through the axon [30].

The part of the axon that has lost contact with the neuronal body is destroyed, and its myelin is phagocytosed by Schwann cells and macrophages. The whole process is known as Wallerian degeneration [26]. As a result of the process, the muscle fibres atrophy very rapidly in the absence of nerve stimuli, and are irreversibly damaged by 18 months. They are then replaced by fatty and fibrous tissue [31,32].

## 4. Clinical

When it comes to sciatic neuropathy, the clinical picture is usually more frequently seen with a lesion at the level of the common division of the fibula than at the tibial division. The common division of the fibula, compared to the tibial division, has fewer and larger fascicles and has less supporting tissue, and is therefore thought to be more vulnerable to compression. In addition, the common division of the fibula is tighter and more secure at the sciatic notch and the neck of the fibula. This makes it potentially more prone to stretch injury [33,34,35].

Clinically, EPSN neuropathy is manifested by weakness in dorsiflexion and eversion of the foot, often causing the person to stub their toe when walking. Inversion of the foot and plantar flexion must be preserved. The toes cannot be extended, with the flexors predominating and causing claw foot [21,36,37,38].

The onset of symptoms varies depending on the cause and extent of the injuries. It may appear abruptly or progressively and start with one symptom or another without the onset of other symptoms [39]. If the lesion is irritative rather than destructive, there may be neuropathic pain, which increases at night, with activity and stretching. Possible symptoms include the Valleix phenomenon, wherein the nerve is sensitive to palpation. Also positivity to the Tinel test, wherein a sensation of electrical discharge is felt along the nerve pathway due to direct percussion on it. In addition, vegetative changes may appear in the autonomic territory of the injured nerve [40].

## 5. Diagnosis

Patient history and clinical examination are key in the diagnosis of nerve entrapment. The examination should include a provocative sign using Tinel’s sign and/or nerve blocks. Peripheral nerve injuries are initially assessed according to the crush dynamics of the nerve injury. During the clinical examination, motor power, sensation, and autonomic nerve functions are also explored [35,40].

When approaching the diagnosis, we should start with electromyography (EMG) [4,41,42,43]. Nerve conduction studies and EMG can identify axonal injury but cannot precisely localise the site of nerve injury. EMG is not valid until 3 weeks have passed and Wallerian degeneration has occurred [35].

Magnetic resonance neurography (MRN) is a new technique to detect peripheral nerve lesions [44,45], and can visualise nerve lesions even at the fascicular microstructural level [46].

The echography offers a less expensive and non-invasive option to guide treatment. Unlike electrodiagnostic studies alone, ultrasound can detect anatomical causes such as scarring, lesions, infiltration of bony fragments, and motion-tethered nerves. Contralateral comparison is often helpful for determining the type of lesion [39,47].

## 6. Treatment

In medicine, there is often a tendency to think that nerve injuries cannot be repaired, but fortunately, this is not the case. This may be due to confusion between the central nervous system (CNS) and the peripheral nervous system (PNS) injuries. Currently, CNS lesions cannot regenerate with surgical intervention; however, PNS lesions can regenerate after intervention. Moreover, it is currently not only limited to nerve regeneration by surgery. In addition, experimental molecular and bioengineering strategies are currently being developed to overcome nerve regeneration and recovery in patients [21,34].

### 6.1. Surgical Treatment

First, when the intrinsic compressive etiology is demonstrated by a space-occupying lesion, for example, by MRN, the lesion will be removed and analysed histologically, as shown in Figure 2A,B.

Another procedure is neurolysis, but its problem is that there is a risk of damaging undamaged nerve bundles. External neurolysis consists of freeing the nerve from its scar environment and involves fibrosis of the epineurium and the elements surrounding the nerve. Internal neurolysis requires the release of compressed fascicular groups, but the risk of injury to the inter-fascicular communications is significant. Both of these are shown in Figure 3A,B.

Grafts are indicated in cases where a suture would be under too much tension or where there is loss of substance between the two ends of the nerve [48,49]. It is always an autograft that must be revascularized from the tissue in which it is placed. This is why only small-diameter nerves can be used, as if the nerve is too thick, it will necrose in the central part.

Another surgical technique is neurotisation, which consists of driving a healthy nerve into a denervated nerve or territory when suturing or grafting is impossible. It is mainly used in brachial plexus injuries, as the absence of healthy donor nerves forces the use of nerves from adjacent regions. In the case of peroneal nerve involvement, transpositions of posterior tibial branches and functional branches of the superficial peroneal nerve, to the deep branch of the peroneal nerve, are performed, as well as innervation of the tibialis anterior muscle, with favourable results in motor recovery [50,51,52].

### 6.2. Biological Treatment

There is now a growing body of research into various cellular, molecular, and bioengineering strategies to promote the repair and recovery of nerve damage. A number of technologies are now available that may help to improve the treatment of peripheral nerve injuries. Their use as an adjunct to surgical nerve repair may help to address the biological limitations of nerve regeneration. Many of the analgesic therapy clinical trials do not discriminate about the type of pain they treat. Cell therapies have emerged as promising potential therapeutics in both spinal cord regeneration and central neuropathic pain mitigation, but most clinical trials are animal-based.

In our opinion, biological techniques could be a paradigm shift in treatment and prognosis after peripheral nerve injury. In principle, we have to consider that nerve fibres regenerate spontaneously, depending on the size of the condition, the neuroma, and the formation of scar tissue [53].

Mesenchymal stem cells (MSCs) used for injuries of the musculoskeletal system can be obtained from the patient’s fat, bone marrow, or even the umbilical cord. Although the mechanism of action is not yet fully elucidated, the use of MSCs is based on their anti-inflammatory property, which can lead to a decrease in the inflammatory response [54].

We must consider that it is possible that peripheral nerves could also be regenerated by the ability of peripheral neurons and Schwann cells (SC) to stimulate appropriate growth [39,55,56]. Early in development, neural crest cells produce SC precursors and other cells (neuronal and non-neuronal) [57,58,59].

In the repair process, Schwann cells provide the signals necessary for the survival and adaptation of injured neurons, axonal regeneration, and reinnervation. Conversion into reparative Schwann cells involves cell dedifferentiation and activation [55]. In this regard, it has already been described that SCs play a key role in the regeneration of axons in peripheral nerve grafts. With myelination, Schwann cells organize themselves. Many axons are introduced deeply into the cellular grafts, but not the acellular peripheral nerve grafts [59,60,61].

Nowadays, the use of biological techniques to deliver growth factors after nerve injury, which also promotes the reprogramming of Schwann cells, can accelerate this regeneration rate [55,62]. Likewise, in our group and in reference to vertebral disc regeneration, we used autologous and allogeneic mesenchymal stromal cells (MSC), which demonstrated their viability, safety, and strong indications of clinical efficacy one year after cell transplantation in the treatment of vertebral discs. They appear to be a valid alternative for the treatment of degenerative disc disease, as they can provide effective and long-lasting pain relief [63,64].

Following nerve injury, Schwann cells are reprogrammed, which involves the activation of repair-supporting elements, including macrophage recruitment, increased cytokines, increased trophic factors, and the removal of destroyed myelin through the autophagic capacity of the SCs themselves [65].

Studies on the importance and utility of stem cells have reported that bone marrow stem cells (BMSCs) have the capacity to differentiate into neuronal lines. These include SC-like cells, astrocytes, and oligodendrocytes [66,67]. It has been reported [68] that BMSCs can restore peripheral nerves through neutrophilic elements, and indirectly by altering SCs [68,69].

Neurotrophins (nerve growth factors) are released from the nerve ending during the process of nerve regeneration. They are released especially after nerve injury and promote nerve differentiation and growth [70].

After nerve injury, the production of nerve growth factor (NGF) is stimulated and plays a key role in the survival of sensory neurons [52]. Other growth factors, such as glial growth factor (GGF), glial cell-derived neurotrophic factor (GDNF), fibroblast growth factor (FGF), neurotrophin-3 (NT-3), ciliary neurotrophic factor, and leupeptin, are also produced in nerve regeneration [53,62,71,72,73].

In addition, other neurotrophic factors and surface proteins that promote axonal elongation include artemin, brain-derived neurotrophic factor (BDNF), neurotrophin-3 (NT-3), vascular endothelial growth factor (VEGF), erythropoietin, pleiotrophin, p75NTR, and N-cadherin [32,63,70,74,75].

In a study from Nath et al. [50], NGF, GGF, GDNF, and NT-3 were applied to small animal models of nerve gap injuries. These authors observed clear histological and electrophysiological improvements [51]. A study comparing NGF-seeded conduits with nerve autografts demonstrated high functional results in the autograft group. This confirms that the application of growth factors in this type of pathology could further enhance axonal regeneration [76].

Moreover, adipose-derived stem cells (ASCs) have been reported to be able to differentiate into cell types contained in different germ layers [75,77] and can effectively support nerve repair [6,78,79,80].

ASCs increased the regeneration and proliferation of proliferating Schwann cells. Indeed, Kingham et al. [81] observed that treatment of ASCs with a combination of mitogenic and distinct elements led to the expression of the glial cell markers S100B, glial fibrillary acidic protein, and the neurotrophic receptor p75 [82,83,84,85,86,87].

The compounds used in Kingham’s induction protocol have different biological functions. For example, forskolin activates adenylyl cyclase, which causes an increase in the level of intracellular cyclic adenosine monophosphate (cAMP), which promotes and enhances the mitogenic responses of SCs [72], in response to that of the growth factors of PDGF and bFGF/FGF2 [78]. Neuregulin-1 (NRG1) is also involved in SC development and progression. It determines the differentiation of Schwann cells into myelinating or non-myelinating cells. This NRG1 generates a cascade of events that promotes SC differentiation and expansion. NRG1 levels will determine axon size, allowing myelinating Schwann cells to optimise myelin sheath thickness [79,80].

Furthermore, ASCs have yielded positive results in studies on a large number of peripheral nerve lesions [81,83], despite uncertainty about their precise dynamics. It is likely that ASCs produce an excess of growth factors, which are critical in the functioning of the peripheral nervous system [87,88].

As indicated by Mathot et al. [6] in a recent review, these adipose tissue-derived mesenchymal stem cells (MSCs) produced by ASCs can successfully differentiate into Schwann-like cells, with the potential to enhance peripheral nerve repair/reconstruction.

## 7. Conclusions

Compression injuries of the EPSN are the most frequent in the lower limb. In this review we highlighted the importance and relevance that biological treatment can have as a complement to traditional surgical treatments. The application of biological growth factors will undoubtedly help to achieve stable nerve recovery. Future research should examine the specific pain response.

## Figures and Tables

**Figure 1 jcm-11-02804-f001:**
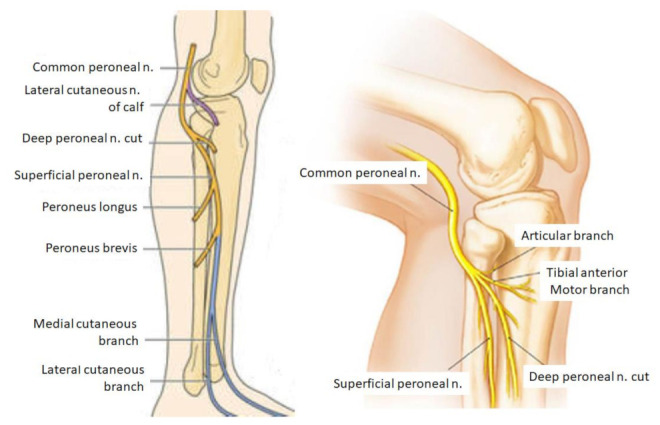
Anatomical presentation of the bifurcations of the sciatic nerve in the popliteal fossa.

**Figure 2 jcm-11-02804-f002:**
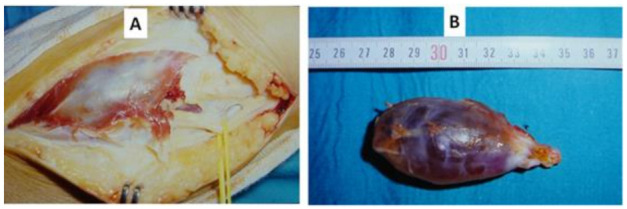
(**A**) Intraoperative image of upper tibio-peroneal cyst compressing the EPSN. (**B**) Sample of the superior tibio-peroneal cyst compressing the EPSN.

**Figure 3 jcm-11-02804-f003:**
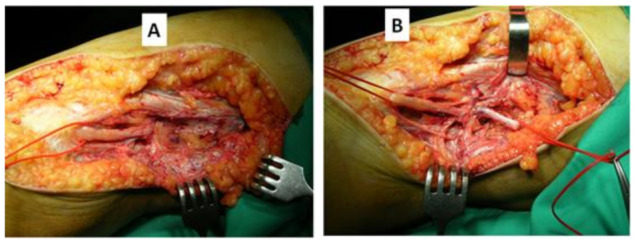
(**A**) Intraoperative image of fibrosis encompassing the EPSN. (**B**) Intraoperative image of the NPCE after neurolysis; note the macroscopic appearance of the nerve in its compressed section.

## Data Availability

Not applicable.

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
