# Peer review of "Biological Approach in the Treatment of External Popliteal Sciatic Nerve (Epsn) Neurological Injury: Review"

_jcm, 2022, doi:10.3390/jcm11102804_

Round 1

Reviewer 1 Report

In this review, the authors elaborate on the biological approach for EPSN compressive etiology. The manuscript is very well written and provides a useful overview about this interesting topic.

A minor point is the outdated literature for Schwann cell biology after injury, Schwann cell (SC) field speaks of repair SC rather than repair SC. 

Author Response

In this review, the authors elaborate on the biological approach for EPSN compressive etiology. The manuscript is very well written and provides a useful overview about this interesting topic.

A minor point is the outdated literature for Schwann cell biology after injury, Schwann cell (SC) field speaks of repair SC rather than repair SC.

In this regard, it has already been described that SCs play a key role in the regeneration of axons in peripheral nerve grafts. With myelination, Schwann cells organise themselves. Many axons are introduced deeply into the cellular grafts but not the acellular peripheral nerve grafts. (Fu et al 2022, Berry et al. (1988).

Reviewer 2 Report

This review describes the novelty of the biological approach in the treatment of external popliteal sciatic nerve. The strength of the paper is to describe the difficult and controversial issue for the orthopedic surgeon and a challenge for rehabilitation and sport medicine field. Actually, there are few publications related to this subject but in this narrative review you deal with the neuropathy of the external popliteal sciatic nerve, including an anatomical reminder, clinical presentation and diagnosis, surgical approach and stressing the new role of biological approach. This last is a quite unexplored field in literature that needs more evidence-based studies, to bring it to the real practitioner world.

Your review is well summarized and the list of references adequate and wide enough, but there are some minor comments to address:

Title

for the first time in the paper the acronym of EPSN appear here, it’s not necessary to repeat it in every chapter between parentheses. You should re-read the test considering and substituting “external popliteal sciatic nerve” with “EPSN”, when necessary.

Abstract and text

Line 29: I think is better to change in “in the treatment of this situation”, cutting out “reversal”

Line 40: In this part could be useful to stress the epidemiology of this kind of injuries (work/sport related? Which age involved? How much converted in disability?). You can link with line 53’s concept.

Lines 47-51: You should better explain the mechanism of injury, conduction blocks and the causes of nerve involvement in the lower extremity. Consider that you should make the reader better understand the gradient of severity of conduction blocks.

Line 60: EPSN not EPCN

Line 72: If possible, you can add in this position a Flow diagram of your narrative review of literature.

Line 83: better to change the title “compression of the EPSN” in something that suggest the etiology and pathophysiology of the lesion, “compression” is the mechanism of injury only.

Line85: headline what you want to focus on.

Figure 1. is it made by you? If not, you should write down the reference.

Line 139- 146: I think you should better explain the issue, without saying “in general medicine, even in some specialties, there is the tendency to think that the nerve injuries cannot be repaired”, it sounds a little bit accusative. You should try to present the treatment’s option as an opportunity.

Line 184-189: Maybe it’s better to move future research at the end of the chapter or in conclusion

Line 269: ESPN

Line 273: it is not clear what do you mean with patient evolution, rehabilitation?

Author Response

This review describes the novelty of the biological approach in the treatment of external popliteal sciatic nerve. The strength of the paper is to describe the difficult and controversial issue for the orthopedic surgeon and a challenge for rehabilitation and sport medicine field. Actually, there are few publications related to this subject but in this narrative review you deal with the neuropathy of the external popliteal sciatic nerve, including an anatomical reminder, clinical presentation and diagnosis, surgical approach and stressing the new role of biological approach. This last is a quite unexplored field in literature that needs more evidence-based studies, to bring it to the real practitioner world.

Your review is well summarized and the list of references adequate and wide enough, but there are some minor comments to address:

 Title

For the first time in the paper the acronym of EPSN appear here, it’s not necessary to repeat it in every chapter between parentheses. You should re-read the test considering and substituting “external popliteal sciatic nerve” with “EPSN”, when necessary.

This comments have been modified

Abstract and text

Line 29: I think is better to change in “in the treatment of this situation”, cutting out “reversal”

Have been modified

Line 40: In this part could be useful to stress the epidemiology of this kind of injuries (work/sport related? Which age involved? How much converted in disability?). You can link with line 53’s concept.

Fibular or peroneal neuropathy is the most common lower limb neuropathy and the third most common focal neuropathy found in general, after median and ulnar neuropa-thies (4). Following high tibial and fibular osteotomies, an incidence of peroneal neuropa-thy has been observed in 2-27% (4). Following knee dislocations, common peroneal nerve injury has been observed in 16-40% of patients. In children, peroneal neuropathy of the common peroneal nerve has also been observed to be affected most frequently (59%), fol-lowed by the deep (12%) and superficial (5%) peroneal nerves (5).

Lines 47-51: You should better explain the mechanism of injury, conduction blocks and the causes of nerve involvement in the lower extremity. Consider that you should make the reader better understand the gradient of severity of conduction blocks.

The pathogenesis of these lesions progressing in complexity from neuropraxia (punctual conduction block due to myelin damage, as in compressive neuropathies), axonotmesis (axonal injury, due to crushing or traction, with irreversible damage associated with denervation time of the target muscle but with a favorable prognosis of the nerve), to neurotmesis (complete section of the nerve with destruction of the endoneurial tubes, requires surgical treatment for resolution, appears in penetrating wounds or ischemic processes), (11).

Microscopic techniques have shown that nerve morphology is normal and neuromuscular junctions are maintained in chronic compression lesions. However, the myelin sheath is thinner and degraded, and there is decreased internodal length (the distance between adjacent nodes of Ranvier)(12)

The main mechanical characteristic of the peripheral nerve is the tensile strength with a non-linear behaviour between weight and deformation. Under constant elongation the nerve tension is reduced to 30% in the first 10 minutes and very little more in the next 20 minutes. This relaxation phenomenon (creep) is useful in sutures and nerve grafts, as the remaining tension will be lower after a short time (13).

Regarding the mechanisms thought to lead to compression injuries. From an ana-tomical point of view, the narrowing of the openings causes an increase in pressure at that site, compresses the blood vessels and leads to nerve ischaemia. Another proposed mechanism is that as a result of lower pressure, which decreases venous return, it can lead to venous stasis. Over time this situation can lead to extraneural oedema, with a con-sequent increase in fibrous tissue around the nerve (13).

Line 60: EPSN not EPCN

Have been corrected

Line 72: If possible, you can add in this position a Flow diagram of your narrative review of literature.

As it is not a systematic review, it is very complicated to make a flow diagram.

Line 83: better to change the title “compression of the EPSN” in something that suggest the etiology and pathophysiology of the lesion, “compression” is the mechanism of injury only.

Have been changed

Line85: headline what you want to focus on.

The title of the figure has been revised

Figure 1. is it made by you? If not, you should write down the reference.

The figure is a composition made by us.

Line 139- 146: I think you should better explain the issue, without saying “in general medicine, even in some specialties, there is the tendency to think that the nerve injuries cannot be repaired”, it sounds a little bit accusative. You should try to present the treatment’s option as an opportunity.

In medicine, there is often a tendency to think that nerve injuries cannot be repaired, but fortunately this is not the case.

Line 184-189: Maybe it’s better to move future research at the end of the chapter or in conclusion

The sentence has been moved to conclusions

Line 269: ESPN

Have been corrected

Line 273: it is not clear what do you mean with patient evolution, rehabilitation?

We think this may be better clarified by the new wording of the sentence: The application of biological growth factors will undoubtedly help to achieve a stable nerve recovery. Future research should examine the specific pain response.